# Translating Embeddings for Modeling Multi-relational Data

**Antoine Bordes, Nicolas Usunier, Alberto Garcia-Durán**
Université de Technologie de Compiègne – CNRS
Heudiasyc UMR 7253
Compiègne, France
{bordesan, nusunier, agarciad}@utc.fr

**Jason Weston, Oksana Yakhnenko**
Google
111 8th avenue
New York, NY, USA
{jweston, oksana}@google.com

## Abstract

We consider the problem of embedding entities and relationships of multi-relational data in low-dimensional vector spaces. Our objective is to propose a canonical model which is easy to train, contains a reduced number of parameters and can scale up to very large databases. Hence, we propose TransE, a method which models relationships by interpreting them as translations operating on the low-dimensional embeddings of the entities. Despite its simplicity, this assumption proves to be powerful since extensive experiments show that TransE significantly outperforms state-of-the-art methods in link prediction on two knowledge bases. Besides, it can be successfully trained on a large scale data set with 1M entities, 25k relationships and more than 17M training samples.

## 1 Introduction

Multi-relational data refers to directed graphs whose nodes correspond to *entities* and *edges* of the form (*head*, *label*, *tail*) (denoted $(h, \ell, t)$), each of which indicates that there exists a relationship of name *label* between the entities *head* and *tail*. Models of multi-relational data play a pivotal role in many areas. Examples are social network analysis, where entities are members and edges (relationships) are friendship/social relationship links, recommender systems where entities are users and products and relationships are buying, rating, reviewing or searching for a product, or knowledge bases (KBs) such as Freebase[1], Google Knowledge Graph[2] or GeneOntology[3], where each entity of the KB represents an abstract concept or concrete entity of the world and relationships are predicates that represent facts involving two of them. Our work focuses on modeling multi-relational data from KBs (Wordnet [9] and Freebase [1] in this paper), with the goal of providing an efficient tool to complete them by automatically adding new facts, without requiring extra knowledge.

**Modeling multi-relational data**   In general, the modeling process boils down to extracting local or global connectivity patterns between entities, and prediction is performed by using these patterns to generalize the observed relationship between a specific entity and all others. The notion of locality for a single relationship may be purely structural, such as the friend of my friend is my friend in

social networks, but can also depend on the entities, such as those who liked Star Wars IV also liked Star Wars V, but they may or may not like Titanic. In contrast to single-relational data where ad-hoc but simple modeling assumptions can be made after some descriptive analysis of the data, the difficulty of relational data is that the notion of locality may involve relationships and entities of different types at the same time, so that modeling multi-relational data requires more generic approaches that can choose the appropriate patterns considering all heterogeneous relationships at the same time.

Following the success of user/item clustering or matrix factorization techniques in collaborative filtering to represent non-trivial similarities between the connectivity patterns of entities in single-relational data, most existing methods for multi-relational data have been designed within the framework of relational learning from latent attributes, as pointed out by [6]; that is, by learning and operating on latent representations (or embeddings) of the constituents (entities and relationships). Starting from natural extensions of these approaches to the multi-relational domain such as non-parametric Bayesian extensions of the *stochastic blockmodel* [7, 10, 17] and models based on tensor factorization [5] or collective matrix factorization [13, 11, 12], many of the most recent approaches have focused on increasing the expressivity and the universality of the model in either Bayesian clustering frameworks [15] or energy-based frameworks for learning embeddings of entities in low-dimensional spaces [3, 15, 2, 14]. The greater expressivity of these models comes at the expense of substantial increases in model complexity which results in modeling assumptions that are hard to interpret, and in higher computational costs. Besides, such approaches are potentially subject to either overfitting since proper regularization of such high-capacity models is hard to design, or underfitting due to the non-convex optimization problems with many local minima that need to be solved to train them. As a matter of fact, it was shown in [2] that a simpler model (linear instead of bilinear) achieves almost as good performance as the most expressive models on several multi-relational data sets with a relatively large number of different relationships. This suggests that even in complex and heterogeneous multi-relational domains simple yet appropriate modeling assumptions can lead to better trade-offs between accuracy and scalability.

**Relationships as translations in the embedding space**   In this paper, we introduce TransE, an energy-based model for learning low-dimensional embeddings of entities. In TransE, relationships are represented as *translations in the embedding space*: if $(h, \ell, t)$ holds, then the embedding of the tail entity $t$ should be close to the embedding of the head entity $h$ plus some vector that depends on the relationship $\ell$. Our approach relies on a reduced set of parameters as it learns only one low-dimensional vector for each entity and each relationship.

The main motivation behind our translation-based parameterization is that hierarchical relationships are extremely common in KBs and translations are the natural transformations for representing them. Indeed, considering the natural representation of trees (i.e. embeddings of the nodes in dimension 2), the siblings are close to each other and nodes at a given height are organized on the $x$-axis, the parent-child relationship corresponds to a translation on the $y$-axis. Since a null translation vector corresponds to an equivalence relationship between entities, the model can then represent the sibling relationship as well. Hence, we chose to use our parameter budget per relationship (one low-dimensional vector) to represent what we considered to be the key relationships in KBs. Another, secondary, motivation comes from the recent work of [8], in which the authors learn word embeddings from free text, and some *1-to-1* relationships between entities of different types, such "capital of" between countries and cities, are (coincidentally rather than willingly) represented by the model as translations in the embedding space. This suggests that there may exist embedding spaces in which *1-to-1* relationships between entities of different types may, as well, be represented by translations. The intention of our model is to enforce such a structure of the embedding space.

Our experiments in Section 4 demonstrate that this new model, despite its simplicity and its architecture primarily designed for modeling hierarchies, ends up being powerful on most kinds of relationships, and can significantly outperform state-of-the-art methods in link prediction on real-world KBs. Besides, its light parameterization allows it to be successfully trained on a large scale split of Freebase containing 1M entities, 25k relationships and more than 17M training samples.

In the remainder of the paper, we describe our model in Section 2 and discuss its connections with related methods in Section 3. We detail an extensive experimental study on Wordnet and Freebase in Section 4, comparing TransE with many methods from the literature. We finally conclude by sketching some future work directions in Section 5.

**Algorithm 1** Learning TransE
***

**input** Training set $S = \{(h, \ell, t)\}$, entities and rel. sets $E$ and $L$, margin $\gamma$, embeddings dim. $k$.

 1: **initialize** $\boldsymbol{\ell} \leftarrow \text{uniform}(-\frac{6}{\sqrt{k}}, \frac{6}{\sqrt{k}})$ for each $\ell \in L$

 2:          $\boldsymbol{\ell} \leftarrow \boldsymbol{\ell}/\left\|\boldsymbol{\ell}\right\|$ for each $\ell \in L$

 3:          $\mathbf{e} \leftarrow \text{uniform}(-\frac{6}{\sqrt{k}}, \frac{6}{\sqrt{k}})$ for each entity $e \in E$

 4: **loop**

 5:     $\mathbf{e} \leftarrow \mathbf{e}/\left\|\mathbf{e}\right\|$ for each entity $e \in E$

 6:     $S_{batch} \leftarrow \text{sample}(S, b)$ // sample a minibatch of size $b$

 7:     $T_{batch} \leftarrow \emptyset$ // initialize the set of pairs of triplets

 8:     **for** $(h, \ell, t) \in S_{batch}$ **do**

 9:        $(h', \ell, t') \leftarrow \text{sample}(S'_{(h,\ell,t)})$ // sample a corrupted triplet

10:        $T_{batch} \leftarrow T_{batch} \cup \left\{ \left( (h, \ell, t), (h', \ell, t') \right) \right\}$

11:     **end for**

12:     Update embeddings w.r.t. $\displaystyle\sum_{\left( (h,\ell,t),(h',\ell,t') \right) \in T_{batch}} \nabla \left[ \gamma + d(\boldsymbol{h} + \boldsymbol{\ell}, \boldsymbol{t}) - d(\boldsymbol{h'} + \boldsymbol{\ell}, \boldsymbol{t'}) \right]_+$

13: **end loop**
***

## 2 Translation-based model

Given a training set $S$ of triplets $(h, \ell, t)$ composed of two entities $h, t \in E$ (the set of entities) and a relationship $\ell \in L$ (the set of relationships), our model learns vector embeddings of the entities and the relationships. The embeddings take values in $\mathbb{R}^k$ ($k$ is a model hyperparameter) and are denoted with the same letters, in boldface characters. The basic idea behind our model is that the functional relation induced by the $\ell$-labeled edges corresponds to a translation of the embeddings, i.e. we want that $\boldsymbol{h} + \boldsymbol{\ell} \approx \boldsymbol{t}$ when $(h, \ell, t)$ holds ($\boldsymbol{t}$ should be a nearest neighbor of $\boldsymbol{h} + \boldsymbol{\ell}$), while $\boldsymbol{h} + \boldsymbol{\ell}$ should be far away from $\boldsymbol{t}$ otherwise. Following an energy-based framework, the energy of a triplet is equal to $d(\boldsymbol{h} + \boldsymbol{\ell}, \boldsymbol{t})$ for some dissimilarity measure $d$, which we take to be either the $L_1$ or the $L_2$-norm.

To learn such embeddings, we minimize a margin-based ranking criterion over the training set:

$$\mathcal{L} = \sum_{(h,\ell,t) \in S} \sum_{(h',\ell,t') \in S'_{(h,\ell,t)}} \left[ \gamma + d(\boldsymbol{h} + \boldsymbol{\ell}, \boldsymbol{t}) - d(\boldsymbol{h'} + \boldsymbol{\ell}, \boldsymbol{t'}) \right]_+ \tag{1}$$

where $[x]_+$ denotes the positive part of $x$, $\gamma > 0$ is a margin hyperparameter, and

$$S'_{(h,\ell,t)} = \left\{ (h', \ell, t) | h' \in E \right\} \cup \left\{ (h, \ell, t') | t' \in E \right\}. \tag{2}$$

The set of corrupted triplets, constructed according to Equation 2, is composed of training triplets with either the head or tail replaced by a random entity (but not both at the same time). The loss function (1) favors lower values of the energy for training triplets than for corrupted triplets, and is thus a natural implementation of the intended criterion. Note that for a given entity, its embedding vector is the same when the entity appears as the head or as the tail of a triplet.

The optimization is carried out by stochastic gradient descent (in minibatch mode), over the possible $\boldsymbol{h}, \boldsymbol{\ell}$ and $\boldsymbol{t}$, with the additional constraints that the $L_2$-norm of the embeddings of the entities is 1 (no regularization or norm constraints are given to the label embeddings $\boldsymbol{\ell}$). This constraint is important for our model, as it is for previous embedding-based methods [3, 6, 2], because it prevents the training process to trivially minimize $\mathcal{L}$ by artificially increasing entity embeddings norms.

The detailed optimization procedure is described in Algorithm 1. All embeddings for entities and relationships are first initialized following the random procedure proposed in [4]. At each main iteration of the algorithm, the embedding vectors of the entities are first normalized. Then, a small set of triplets is sampled from the training set, and will serve as the training triplets of the minibatch. For each such triplet, we then sample a single corrupted triplet. The parameters are then updated by taking a gradient step with constant learning rate. The algorithm is stopped based on its performance on a validation set.

## 3 Related work

Section 1 described a large body of work on embedding KBs. We detail here the links between our model and those of [3] (Structured Embeddings or SE) and [14].

Table 1: **Numbers of parameters** and their values for FB15k (in millions). $n_e$ and $n_r$ are the nb. of entities and relationships; $k$ the embeddings dimension.

| METHOD | NB. OF PARAMETERS | ON FB15K |
|---|---|---|
| Unstructured [2] | $O(n_e k)$ | 0.75 |
| RESCAL [11] | $O(n_e k + n_r k^2)$ | 87.80 |
| SE [3] | $O(n_e k + 2 n_r k^2)$ | 7.47 |
| SME(LINEAR) [2] | $O(n_e k + n_r k + 4 k^2)$ | 0.82 |
| SME(BILINEAR) [2] | $O(n_e k + n_r k + 2 k^3)$ | 1.06 |
| LFM [6] | $O(n_e k + n_r k + 10 k^2)$ | 0.84 |
| TransE | $O(n_e k + n_r k)$ | 0.81 |

Table 2: **Statistics of the data sets** used in this paper and extracted from the two knowledge bases, Wordnet and Freebase.

| DATA SET | WN | FB15K | FB1M |
|---|---|---|---|
| ENTITIES | 40,943 | 14,951 | $1 \times 10^6$ |
| RELATIONSHIPS | 18 | 1,345 | 23,382 |
| TRAIN. EX. | 141,442 | 483,142 | $17.5 \times 10^6$ |
| VALID EX. | 5,000 | 50,000 | 50,000 |
| TEST EX. | 5,000 | 59,071 | 177,404 |

SE [3] embeds entities into $\mathbb{R}^k$, and relationships into two matrices $\boldsymbol{L}_1 \in \mathbb{R}^{k \times k}$ and $\boldsymbol{L}_2 \in \mathbb{R}^{k \times k}$ such that $d(\boldsymbol{L}_1 \boldsymbol{h}, \boldsymbol{L}_2 \boldsymbol{t})$ is large for corrupted triplets $(h, \ell, t)$ (and small otherwise). The basic idea is that when two entities belong to the same triplet, their embeddings should be close to each other in some subspace that depends on the relationship. Using two different projection matrices for the head and for the tail is intended to account for the possible asymmetry of relationship $\ell$. When the dissimilarity function takes the form of $d(\boldsymbol{x}, \boldsymbol{y}) = g(\boldsymbol{x} - \boldsymbol{y})$ for some $g : \mathbb{R}^k \to \mathbb{R}$ (e.g. $g$ is a norm), then SE with an embedding of size $k + 1$ is strictly more expressive than our model with an embedding of size $k$, since linear operators in dimension $k + 1$ can reproduce affine transformations in a subspace of dimension $k$ (by constraining the $k+1$th dimension of all embeddings to be equal to 1). SE, with $\boldsymbol{L}_2$ as the identity matrix and $\boldsymbol{L}_1$ taken so as to reproduce a translation is then equivalent to TransE. Despite the lower expressiveness of our model, we still reach better performance than SE in our experiments. We believe this is because (1) our model is a more direct way to represent the true properties of the relationship, and (2) optimization is difficult in embedding models. For SE, greater expressiveness seems to be more synonymous to underfitting than to better performance. Training errors (in Section 4.3) tend to confirm this point.

Another related approach is the Neural Tensor Model [14]. A special case of this model corresponds to learning scores $s(h, \ell, t)$ (lower scores for corrupted triplets) of the form:

$$s(h, \ell, t) = \boldsymbol{h}^T \boldsymbol{L} \boldsymbol{t} + \boldsymbol{\ell}_1^T \boldsymbol{h} + \boldsymbol{\ell}_2^T \boldsymbol{t} \tag{3}$$

where $\boldsymbol{L} \in \mathbb{R}^{k \times k}$, $\boldsymbol{L}_1 \in \mathbb{R}^k$ and $\boldsymbol{L}_2 \in \mathbb{R}^k$, all of them depending on $\ell$.

If we consider TransE with the squared euclidean distance as dissimilarity function, we have:

$$d(\boldsymbol{h} + \boldsymbol{\ell}, \boldsymbol{t}) = \| \boldsymbol{h} \|_2^2 + \| \boldsymbol{\ell} \|_2^2 + \| \boldsymbol{t} \|_2^2 - 2 (\boldsymbol{h}^T \boldsymbol{t} + \boldsymbol{\ell}^T (\boldsymbol{t} - \boldsymbol{h})) .$$

Considering our norm constraints ($\| \boldsymbol{h} \|_2^2 = \| \boldsymbol{t} \|_2^2 = 1$) and the ranking criterion (1), in which $\| \boldsymbol{\ell} \|_2^2$ does not play any role in comparing corrupted triplets, our model thus involves scoring the triplets with $\boldsymbol{h}^T \boldsymbol{t} + \boldsymbol{\ell}^T (\boldsymbol{t} - \boldsymbol{h})$, and hence corresponds to the model of [14] (Equation (3)) where $\boldsymbol{L}$ is the identity matrix, and $\boldsymbol{\ell} = \boldsymbol{\ell}_1 = -\boldsymbol{\ell}_2$. We could not run experiments with this model (since it has been published simultaneously as ours), but once again TransE has much fewer parameters: this could simplify the training and prevent underfitting, and may compensate for a lower expressiveness.

Nevertheless, the simple formulation of TransE, which can be seen as encoding a series of 2-way interactions (e.g. by developing the $L_2$ version), involves drawbacks. For modeling data where 3-way dependencies between $h$, $\ell$ and $t$ are crucial, our model can fail. For instance, on the small-scale Kinships data set [7], TransE does not achieve performance in cross-validation (measured with the area under the precision-recall curve) competitive with the state-of-the-art [11, 6], because such ternary interactions are crucial in this case (see discussion in [2]). Still, our experiments of Section 4 demonstrate that, for handling generic large-scale KBs like Freebase, one should first model properly the most frequent connectivity patterns, as TransE does.

## 4 Experiments

Our approach, TransE, is evaluated on data extracted from Wordnet and Freebase (their statistics are given in Table 2), against several recent methods from the literature which were shown to achieve the best current performance on various benchmarks and to scale to relatively large data sets.

### 4.1 Data sets

**Wordnet** This KB is designed to produce an intuitively usable dictionary and thesaurus, and support automatic text analysis. Its entities (termed *synsets*) correspond to word senses, and relationships define lexical relations between them. We considered the data version used in [2], which we denote WN in the following. Examples of triplets are (_score_NN_1, _hypernym, _evaluation_NN_1) or (_score_NN_2, _has_part, _musical_notation_NN_1).[4]

**Freebase** Freebase is a huge and growing KB of general facts; there are currently around 1.2 billion triplets and more than 80 million entities. We created two data sets with Freebase. First, to make a small data set to experiment on we selected the subset of entities that are also present in the Wikilinks database[5] and that also have at least 100 mentions in Freebase (for both entities and relationships). We also removed relationships like '!/people/person/nationality' which just reverses the head and tail compared to the relationship '/people/person/nationality'. This resulted in 592,213 triplets with 14,951 entities and 1,345 relationships which were randomly split as shown in Table 2. This data set is denoted *FB15k* in the rest of this section. We also wanted to have large-scale data in order to test TransE at scale. Hence, we created another data set from Freebase, by selecting the most frequently occurring 1 million entities. This led to a split with around 25k relationships and more than 17 millions training triplets, which we refer to as *FB1M*.

### 4.2 Experimental setup

**Evaluation protocol** For evaluation, we use the same ranking procedure as in [3]. For each test triplet, the head is removed and replaced by each of the entities of the dictionary in turn. Dissimilarities (or energies) of those corrupted triplets are first computed by the models and then sorted by ascending order; the rank of the correct entity is finally stored. This whole procedure is repeated while removing the tail instead of the head. We report the *mean* of those predicted ranks and the *hits@10*, i.e. the proportion of correct entities ranked in the top 10.

These metrics are indicative but can be flawed when some corrupted triplets end up being valid ones, from the training set for instance. In this case, those may be ranked above the test triplet, but this should not be counted as an error because both triplets are true. To avoid such a misleading behavior, we propose to remove from the list of corrupted triplets all the triplets that appear either in the training, validation or test set (except the test triplet of interest). This ensures that all corrupted triplets do not belong to the data set. In the following, we report mean ranks and hits@10 according to both settings: the original (possibly flawed) one is termed *raw*, while we refer to the newer as *filtered* (or *filt.*). We only provide *raw* results for experiments on FB1M.

**Baselines** The first method is Unstructured, a version of TransE which considers the data as mono-relational and sets all translations to **0** (it was already used as baseline in [2]). We also compare with RESCAL, the collective matrix factorization model presented in [11, 12], and the energy-based models SE [3], SME(linear)/SME(bilinear) [2] and LFM [6]. RESCAL is trained via an alternating least-square method, whereas the others are trained by stochastic gradient descent, like TransE. Table 1 compares the theoretical number of parameters of the baselines to our model, and gives the order of magnitude on FB15k. While SME(linear), SME(bilinear), LFM and TransE have about the same number of parameters as Unstructured for low dimensional embeddings, the other algorithms SE and RESCAL, which learn at least one $k \times k$ matrix for each relationship rapidly need to learn many parameters. RESCAL needs about $87$ times more parameters on FB15k because it requires a much larger embedding space than other models to achieve good performance. We did not experiment on FB1M with RESCAL, SME(bilinear) and LFM for scalability reasons in terms of numbers of parameters or training duration.

We trained all baseline methods using the code provided by the authors. For RESCAL, we had to set the regularization parameter to $0$ for scalability reasons, as it is indicated in [11], and chose the latent dimension $k$ among $\{50, 250, 500, 1000, 2000\}$ that led to the lowest mean predicted ranks on the validation sets (using the *raw* setting). For Unstructured, SE, SME(linear) and SME(bilinear), we

Table 3: **Link prediction results.** Test performance of the different methods.

| DATASET | WN | | | | FB15K | | | | FB1M | |
|---|---|---|---|---|---|---|---|---|---|---|
| METRIC | MEAN RANK | | HITS@10 (%) | | MEAN RANK | | HITS@10 (%) | | MEAN RANK | HITS@10 (%) |
| *Eval. setting* | *Raw* | *Filt.* | *Raw* | *Filt.* | *Raw* | *Filt.* | *Raw* | *Filt.* | *Raw* | *Raw* |
| Unstructured [2] | 315 | 304 | 35.3 | 38.2 | 1,074 | 979 | 4.5 | 6.3 | 15,139 | 2.9 |
| RESCAL [11] | 1,180 | 1,163 | 37.2 | 52.8 | 828 | 683 | 28.4 | 44.1 | - | - |
| SE [3] | 1,011 | 985 | 68.5 | 80.5 | 273 | 162 | 28.8 | 39.8 | 22,044 | 17.5 |
| SME(LINEAR) [2] | 545 | 533 | 65.1 | 74.1 | 274 | 154 | 30.7 | 40.8 | - | - |
| SME(BILINEAR) [2] | 526 | 509 | 54.7 | 61.3 | 284 | 158 | 31.3 | 41.3 | - | - |
| LFM [6] | 469 | 456 | 71.4 | 81.6 | 283 | 164 | 26.0 | 33.1 | - | - |
| TransE | **263** | **251** | **75.4** | **89.2** | **243** | **125** | **34.9** | **47.1** | **14,615** | **34.0** |

selected the learning rate among $\{0.001, 0.01, 0.1\}$, $k$ among $\{20, 50\}$, and selected the best model by early stopping using the mean rank on the validation sets (with a total of at most 1,000 epochs over the training data). For LFM, we also used the mean validation ranks to select the model and to choose the latent dimension among $\{25, 50, 75\}$, the number of factors among $\{50, 100, 200, 500\}$ and the learning rate among $\{0.01, 0.1, 0.5\}$.

**Implementation**    For experiments with TransE, we selected the learning rate $\lambda$ for the stochastic gradient descent among $\{0.001, 0.01, 0.1\}$, the margin $\gamma$ among $\{1, 2, 10\}$ and the latent dimension $k$ among $\{20, 50\}$ on the validation set of each data set. The dissimilarity measure $d$ was set either to the $L_1$ or $L_2$ distance according to validation performance as well. Optimal configurations were: $k = 20$, $\lambda = 0.01$, $\gamma = 2$, and $d = L_1$ on Wordnet; $k = 50$, $\lambda = 0.01$, $\gamma = 1$, and $d = L_1$ on FB15k; $k = 50$, $\lambda = 0.01$, $\gamma = 1$, and $d = L_2$ on FB1M. For all data sets, training time was limited to at most $1,000$ epochs over the training set. The best models were selected by early stopping using the mean predicted ranks on the validation sets (*raw* setting). An open-source implementation of TransE is available from the project webpage[6].

## 4.3    Link prediction

**Overall results**    Tables 3 displays the results on all data sets for all compared methods. As expected, the *filtered* setting provides lower mean ranks and higher hits@10, which we believe are a clearer evaluation of the performance of the methods in link prediction. However, generally the trends between *raw* and *filtered* are the same.

Our method, TransE, outperforms all counterparts on all metrics, usually with a wide margin, and reaches some promising absolute performance scores such as 89% of hits@10 on WN (over more than 40k entities) and 34% on FB1M (over 1M entities). All differences between TransE and the best runner-up methods are important.

We believe that the good performance of TransE is due to an appropriate design of the model according to the data, but also to its relative simplicity. This means that it can be optimized efficiently with stochastic gradient. We showed in Section 3 that SE is more expressive than our proposal. However, its complexity may make it quite hard to learn, resulting in worse performance. On FB15k, SE achieves a mean rank of 165 and hits@10 of 35.5% on a subset of 50k triplets of the training set, whereas TransE reaches 127 and 42.7%, indicating that TransE is indeed less subject to underfitting and that this could explain its better performances. SME(bilinear) and LFM suffer from the same training issue: we never managed to train them well enough so that they could exploit their full capabilities. The poor results of LFM might also be explained by our evaluation setting, based on ranking entities, whereas LFM was originally proposed to predict relationships. RESCAL can achieve quite good hits@10 on FB15k but yields poor mean ranks, especially on WN, even when we used large latent dimensions ($2,000$ on Wordnet).

The impact of the translation term is huge. When one compares performance of TransE and Unstructured (i.e. TransE without translation), mean ranks of Unstructured appear to be rather good (best runner-up on WN), but hits@10 are very poor. Unstructured simply clusters all entities co-occurring together, independent of the relationships involved, and hence can only make guesses of which entities are related. On FB1M, the mean ranks of TransE and Unstructured are almost similar, but TransE places 10 times more predictions in the top 10.

Table 4: **Detailed results by category of relationship.** We compare Hits@10 (in %) on FB15k in the filtered evaluation setting for our model, TransE and baselines. (M. stands for MANY).

| TASK | PREDICTING *head* | | | | PREDICTING *tail* | | | |
|---|---|---|---|---|---|---|---|---|
| REL. CATEGORY | 1-TO-1 | 1-TO-M. | M.-TO-1 | M.-TO-M. | 1-TO-1 | 1-TO-M. | M.-TO-1 | M.-TO-M. |
| Unstructured [2] | 34.5 | 2.5 | 6.1 | 6.6 | 34.3 | 4.2 | 1.9 | 6.6 |
| SE [3] | 35.6 | 62.6 | 17.2 | 37.5 | 34.9 | 14.6 | 68.3 | 41.3 |
| SME(LINEAR) [2] | 35.1 | 53.7 | 19.0 | 40.3 | 32.7 | 14.9 | 61.6 | 43.3 |
| SME(BILINEAR) [2] | 30.9 | **69.6** | **19.9** | 38.6 | 28.2 | 13.1 | **76.0** | 41.8 |
| TransE | **43.7** | 65.7 | 18.2 | **47.2** | **43.7** | **19.7** | 66.7 | **50.0** |

Table 5: **Example predictions** on the FB15k test set using TransE. **Bold** indicates the test triplet's true tail and *italics* other true tails present in the training set.

| INPUT (HEAD AND LABEL) | PREDICTED TAILS |
|---|---|
| J. K. Rowling `influenced by` | *G. K. Chesterton*, J. R. R. Tolkien, *C. S. Lewis*, **Lloyd Alexander**, Terry Pratchett, Roald Dahl, Jorge Luis Borges, *Stephen King*, Ian Fleming |
| Anthony LaPaglia `performed in` | *Lantana*, *Summer of Sam*, *Happy Feet*, *The House of Mirth*, Unfaithful, **Legend of the Guardians**, Naked Lunch, X-Men, The Namesake |
| Camden County `adjoins` | **Burlington County**, *Atlantic County*, *Gloucester County*, Union County, Essex County, New Jersey, Passaic County, Ocean County, Bucks County |
| The 40-Year-Old Virgin `nominated for` | *MTV Movie Award for Best Comedic Performance*, *BFCA Critics' Choice Award for Best Comedy*, *MTV Movie Award for Best On-Screen Duo*, MTV Movie Award for Best Breakthrough Performance, **MTV Movie Award for Best Movie**, MTV Movie Award for Best Kiss, D. F. Zanuck Producer of the Year Award in Theatrical Motion Pictures, Screen Actors Guild Award for Best Actor - Motion Picture |
| Costa Rica football team `has position` | *Forward*, *Defender*, *Midfielder*, **Goalkeepers**, Pitchers, Infielder, Outfielder, Center, Defenseman |
| Lil Wayne `born in` | **New Orleans**, Atlanta, Austin, St. Louis, Toronto, New York City, Wellington, Dallas, Puerto Rico |
| WALL-E `has the genre` | Animations, Computer Animation, *Comedy film*, *Adventure film*, *Science Fiction*, **Fantasy**, Stop motion, *Satire*, Drama |

**Detailed results** Table 4 classifies the results (in hits@10) on FB15k depending on several categories of the relationships and on the argument to predict for several of the methods. We categorized the relationships according to the cardinalities of their *head* and *tail* arguments into four classes: 1-TO-1, 1-TO-MANY, MANY-TO-1, MANY-TO-MANY. A given relationship is 1-TO-1 if a *head* can appear with at most one *tail*, 1-TO-MANY if a *head* can appear with many *tails*, MANY-TO-1 if many *heads* can appear with the same *tail*, or MANY-TO-MANY if multiple *heads* can appear with multiple *tails*. We classified the relationships into these four classes by computing, for each relationship $\ell$, the averaged number of *heads* $h$ (respect. *tails* $t$) appearing in the FB15k data set, given a pair $(\ell, t)$ (respect. a pair $(h, \ell)$). If this average number was below 1.5 then the argument was labeled as 1 and MANY otherwise. For example, a relationship having an average of 1.2 *head* per *tail* and of 3.2 *tails* per *head* was classified as *1-to-Many*. We obtained that FB15k has 26.2% of 1-TO-1 relationships, 22.7% of 1-TO-MANY, 28.3% of MANY-TO-1, and 22.8% of MANY-TO-MANY.

These detailed results in Table 4 allow for a precise evaluation and understanding of the behavior of the methods. First, it appears that, as one would expect, it is easier to predict entities on the "side 1" of triplets (i.e., predicting *head* in 1-TO-MANY and *tail* in MANY-TO-1), that is when multiple entities point to it. These are the well-posed cases. SME(bilinear) proves to be very accurate in such cases because they are those with the most training examples. Unstructured performs well on 1-TO-1 relationships: this shows that arguments of such relationships must share common hidden types that Unstructured is able to somewhat uncover by clustering entities linked together in the embedding space. But this strategy fails for any other category of relationship. Adding the translation term (i.e. upgrading Unstructured into TransE) brings the ability to move in the embeddings space, from one entity cluster to another by following relationships. This is particularly spectacular for the well-posed cases.

**Illustration** Table 5 gives examples of link prediction results of TransE on the FB15k test set (predicting *tail*). This illustrates the capabilities of our model. Given a head and a label, the top

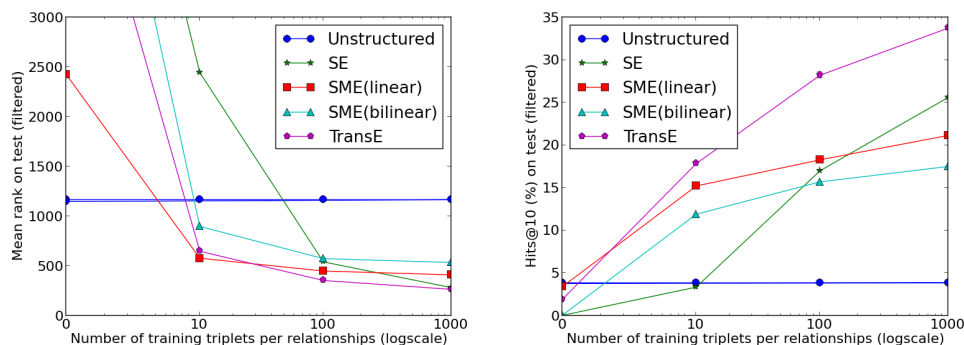

Figure 1: **Learning new relationships with few examples**. Comparative experiments on FB15k data evaluated in mean rank (left) and hits@10 (right). More details in the text.

predicted tails (and the true one) are depicted. The examples come from the FB15k test set. Even if the good answer is not always top-ranked, the predictions reflect common-sense.

### 4.4 Learning to predict new relationships with few examples

Using FB15k, we wanted to test how well methods could generalize to new facts by checking how fast they were learning new relationships. To that end, we randomly selected 40 relationships and split the data into two sets: a set (named *FB15k-40rel*) containing all triplets with these 40 relationships and another set (*FB15k-rest*) containing the rest. We made sure that both sets contained all entities. *FB15k-rest* has then been split into a training set of 353,788 triplets and a validation set of 53,266, and *FB15k-40rel* into a training set of 40,000 triplets (1,000 for each relationship) and a test set of 45,159. Using these data sets, we conducted the following experiment: (1) models were trained and selected using *FB15k-rest* training and validation sets, (2) they were subsequently trained on the training set *FB15k-40rel* but only to learn the parameters related to the fresh 40 relationships, (3) they were evaluated in link prediction on the test set of *FB15k-40rel* (containing only relationships unseen during phase (1)). We repeated this procedure while using 0, 10, 100 and 1000 examples of each relationship in phase (2).

Results for Unstructured, SE, SME(linear), SME(bilinear) and TransE are presented in Figure 1. The performance of Unstructured is the best when no example of the unknown relationship is provided, because it does not use this information to predict. But, of course, this performance does not improve while providing labeled examples. TransE is the fastest method to learn: with only 10 examples of a new relationship, the hits@10 is already 18% and it improves monotonically with the number of provided samples. We believe the simplicity of the TransE model makes it able to generalize well, without having to modify any of the already trained embeddings.

## 5 Conclusion and future work

We proposed a new approach to learn embeddings of KBs, focusing on the minimal parametrization of the model to primarily represent hierarchical relationships. We showed that it works very well compared to competing methods on two different knowledge bases, and is also a highly scalable model, whereby we applied it to a very large-scale chunk of Freebase data. Although it remains unclear to us if all relationship types can be modeled adequately by our approach, by breaking down the evaluation into categories (*1-to-1*, *1-to-Many*, . . . ) it appears to be performing well compared to other approaches across all settings.

Future work could analyze this model further, and also concentrates on exploiting it in more tasks, in particular, applications such as learning word representations inspired by [8]. Combining KBs with text as in [2] is another important direction where our approach could prove useful. Hence, we recently fruitfully inserted TransE into a framework for relation extraction from text [16].

#### Acknowledgments

This work was carried out in the framework of the Labex MS2T (ANR-11-IDEX-0004-02), and funded by the French National Agency for Research (EVEREST-12-JS02-005-01). We thank X. Glorot for providing the code infrastructure, T. Strohmann and K. Murphy for useful discussions.

## Footnotes

[1] freebase.com

[2] google.com/insidesearch/features/search/knowledge.html

[3] geneontology.org

[4]WN is composed of senses, its entities are denoted by the concatenation of a word, its part-of-speech tag and a digit indicating which sense it refers to i.e. _score_NN_1 encodes the first meaning of the noun "score".

[5]`code.google.com/p/wiki-links`

[6]Available at `http://goo.gl/0PpKQe`.

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
