[Reviews · NeurIPS 2013]

Submitted by Assigned_Reviewer_4

The authors propose a simple and scalable approach to modeling multi-relational
data using low-dimensional vector embeddings of entities, with the relationships
between embeddings captured using offset vectors. The embeddings are learned by
training a margin-based ranking model to score the observed
entity1,relationship,entity2 triples higher than the unobserved ones.

Though the proposed model can be seen as a special case of several existing
models (e.g. [1], [2], [3]), it seems to work considerably better than them
despite (or most likely because of) being considerably simpler. The approach
is well motivated and clearly described. The empirical evaluation is reasonably
well done, but the write up could be better. The paper really should report the
hyperparameter values that produced the best performing models along with the
corresponding training times and the parameter counts. It is also unclear
whether L1 or L2 distances were used in the experiments.

Unfortunately, the authors do not provide a clear explanation of the fact that
despite being less expressive than several of the models it is compared to,
TransE outperforms them. It is important to understand whether this is due to
overfitting or underfitting, as answering this question might lead to better
results. Reporting model performance on a subset of training data would be
quite helpful for this. If the more expressive models underfit, TransE can be
used to initialize them better, and if they overfit, TransE can be used to
regularize them in an interesting way.

It might be clearer to define the margin loss in terms of scores instead of
energies, as is more common in the ranking literature.

Why do the numbers for SME do not match those from [2] on WordNet? Was a
different setup used?

Why was the Neural Tensor Model not included in the experimental evaluation?

How important is it to constrain the L2 norm of the embeddings of entities to
be 1?

Typos:
Line 3 of Algorithm 1: j => k
Table 1 caption: d => k
Table 2: 10^9 => 10^6
Line 183: learn=>learning
Summary: A well executed paper that introduces a simple and effective method for
modeling multi-relational data.

Submitted by Assigned_Reviewer_5

The paper describes a simple model denoted TransE for learning embeddings of entities and dyadic relationships between pairs of entities. The TransE model is an important special case of other models that already exist in the literature. The simplifying assumptions of the model enable more effective and rapid training which in turn yield improved results on the two datasets used in this paper over alternative models from the literature. The model operates on triplets (h,l,t) where h is the "head" entity, "l" indicates the type of relation, and "t" is the tail entity. The energy assigned to a triplet (where vect(h) indicates the embedding vector for item h learned by the model) is d(vect(h) + vect(l), vect(t)), where d(.) is either the L1 or L2 distance. The paper describes results on predicting links in two knowledge bases: WordNet and a subset of Freebase. On both datasets TransE substnatially outperforms all of the baselines.

The paper is clear and the organization is straightforward. The choice of the filtered evaluation metric makes sense and the baselines are reasonable. However, I would have appreciated a few more details on the experimental results since the demonstration of the success of TransE is the important contribution of the paper. For example, whether L1 or L2 distance worked better should be included. Since the model is so simple, a more extensive exploration of its failure modes would be valuable. Are there any patterns in the relations or entities it fails to perform well on? Presenting some clearly erroneous examples along with hypotheses about why the model failed to handle those cases would be quite interesting.

What effect does the norm constraint on the embedding vectors have? Do any of the baseline models use a similar constraint? Some alternative models certainly don't use such a constraint and I wonder how important the constraint is to the success of the TransE training procedure and model. A sentence or two on this issue would improve the paper.

Small corrections: The last line of page six should have "Many-To-Many" instead of "1-To-Many."
Summary: This paper is clear and provides a useful empirical examination of an appealing multi-relational embedding learning model that is closely related to other models in the literature. The experiments in the paper justify giving the particular model more attention from the community.

Submitted by Assigned_Reviewer_6

This paper presents a model of multi-relational data, such as the set of facts of the form (subject, predicate, object) that comprise the Freebase database. Perhaps the most common use case of such a model is the following: given a triple with either the subject or object missing, rank all possible completions of that triple. The proposed model is energy-based, and thus assigns a score to each possible completion. The lower the score, the more likely that triple represents a true fact that is typically not already present in the database. In addition to a description of the model, the paper presents a training algorithm and reports large-scale experimental results on Freebase and other datasets on the aforementioned completion task.

The essense of the model is to embed each entity and relation into a continuous vector space. The dimensions of this space have no direct interpretable meaning except that entities which are semantically similar (in the sense of participating in the same set of facts) will be nearby in the latent space (according to L1 or L2), as is standard in matrix factorization. The latent representation of the tail of a triple is modeled as a translation of the latent representation of the head of the triple, with a different learned translation vector for each relation. The quality of a proposed latent embedding is given by an objective function which is essentially the total score of all observed true facts minus the total score of all triples not known to be true, plus a margin term. The parameters of the model (a latent vector for each entity and relation) are learned via a straightforward stochastic gradient descent algorithm from a training set of known facts.

Many similar latent-embedding models of multi-relational data exist in the literature. The main novelty in this paper is the restriction that the embedding of tail of a triple be a relationship-dependent translation of the embedding of the subject, rather than an arbitrary linear transformation. I believe this boils down to the assumption that the contribution of each of the features represented by each dimension of the latent space to the energy function are independent of each other, for a given relationship. As the authors points out, this is strictly a special case of of the 'Structured Embeddings' model. SE embeds each relationship into a matrix rather than a vector, allowing for the possiblity that the transformation represented by the relationship between head and tail embeddings is an arbitrary linear transformation.

The experimental results use the established metrics for this kind of model and show impressive state-of-the-art performance on standard benchmarks. It seems that the expressivity reduction of this model allows for more efficient training and more resilience to overfitting. As these benchmarks are standard, I have no serious complaint with the methodology here. Some minor comments: It would have been interesting to also see results for a task where a missing relationship has to be predicted given a head and tail. The 'filtered' dataset concept, while a good innovation over the 'raw' dataset, seems superflous to this paper given how close the results are between them and since it was not even used for the FB1M dataset, which is the largest and perhaps most important dataset the authors tested on. The authors claim all experimental results are significant (line 302), but do not name the statistical test being used or justify the assumptions behind the test. They also do not say how the latent dimensionality constant was chosen (268).

My main concern with this paper is that is fairly light on novelty and is rather more incremental than a top-tier conference demands. The model is a simple restriction of Structued Embedding, an existing, well-known model: relationship embeddings are replaced with vectors instead of matrices, and so encode translations in latent space rather than arbitrary linear transformations. The learning algorithm is entirely standard stochastic gradient descent. The empirical results are standard benchmarks on standard datasets. While the restriction is shown to be benefial on these tasks, there is somewhat of a lack of in-depth analysis of why. Vague claims in lines 71-93 and on 304 ('appropriate design of the model according to the data') are made that this model is well-suited to the data, but the experiments do not make it clear what is actually helping over SE: Is it that the model is more efficiently trainable? Is it the margin term in the optimization function? Is it that it inherently generalizes better? What exactly is it about these datasets that make translation a good model of relations? The authors present each of these as hypotheses but do not elaborate. As it stands, I feel this paper is presenting a useful idea which could become the core of a good paper, but it needs a deeper analysis of why and when this restriction works.

Some minor comments: * The handling of negative data and the role the closed-world assumption is making to the training procedure was somewhat unclear to me in 134-148. What exactly is being assumed about the truth of facts missing from the database? * On 195, the authors state that experiments with the Neural Tensor Model could not be run, but do not give a reason. * On 236, they state that a corrupted triple 'would' be ranked about the test triplet if the corrupted triple is in fact true, but I would think it may or may not be. * The paper only has a single figure, which comes at the end and displays results. I think a figure earlier in the paper presenting a graphical depiction of the model and its output would be useful.
Summary: The authors present a model and technically sound training algorithm for multi-relational data that achieves state-of-the-art performance on accepted benchmarks. However, the model is a small variation of an existing established model and the training algorithm is standard, calling into question the novelty of the work.

Submitted by Assigned_Reviewer_7

This paper describes a nice method for relation classification using simple vector additions.

The method is very simple and a special case of previous models.
But it was shown that this special case works well even with few parameters.

Added after rebuttal: Upped score to 6.
The data used from [2] is still very problematic since it includes test triplets (x,Relation,y) for many symmetric relationships such as is_similar and with (y,Relation,x) in the training set. I believe that is one of the reasons the simple translation model performs so well. It would be good to exclude these many simple test cases. But since it's hard to verify whether that's the main reason, I upped to score to 6.
Summary: This paper describes a nice and simple method for relation classification using simple vector additions.

The evaluation is ok, some more comparisons to recent work on tensor factorization would have made the paper better.

It seems like this model would work well for relations with very little training data (since the model cannot overfit so easily) but worse for datasets with more triplets to train with. An analysis of this would be nice.
Author Feedback

Author rebuttal: We thank the reviewers for their very detailed and helpful comments. We try to address them below, with a particular focus on detailing the novelty and behavior of the model.

** Novelty/interest **
The main contribution (hence novelty) of our paper consists in the brand new model formulation, which fits nicely the KBs. We believe that its very interesting properties could be of great use for people in the NIPS community interested in modeling/using multi-relational data:
- It is easy to train (less underfitting see below) and to implement. This seems crucial on this problem for which optimization can not be properly conducted.
- It involves less parameters (more compact).
- It performs well on real-world KB data.
- It can be limited to more sophisticated multi-relational data (see discussion below) but, as pointed by R4, it could then serve as initialization for more complex systems, and potentially reduce their underfitting.


** Why does the model work well? **
- R4/R6: "Reporting model performance on a subset of training data would be quite helpful for this" "it that the model is more efficiently trainable? Is it the margin term in the optimization function? Is it that it inherently generalizes better?”

Here are some train results on FB15k (mean rank/hits@10):
- Unstructured : 938 / 5.3%
- SE : 165 / 35.5%
- SME(linear) : 151 / 36.4%
- SME(bilinear) : 155/ 34.2%
- TransE : 127 / 42.7%

This shows that TransE gets the best results on train (as on test), indicating that this model is actually better trained, whereas the other, more expressive, approaches like SE are underfitting. This is not due to the margin term since all above methods also use a margin in training, but to the simple architecture of TransE.

- R7: “It seems like this model would work well for relations with very little training data (since the model cannot overfit so easily) but worse for datasets with more triplets to train with.”

Our model is unlikely to overfit (less parameters), but as we showed above, it is also less subject to underfitting. So, it should work well in both case, large data sets and small data sets. The experiment of Section 4.4 confirms that.

** Failure modes **
R5/R6:"a more extensive exploration of its failure modes would be valuable. Are there any patterns in the relations or entities it fails to perform well on." " What exactly is it about these datasets that make translation a good model of relations?"

The translation-based formulation of TransE can be seen as encoding a series of two-way interactions (e.g. by developing the L2 version). This seems satisfactory to model data from generic KBs, since they do not involve too many relationships with complex ternary interactions. However, for other datasets, for which one must take into account both the 3-way dependencies between h, l and t, TransE should fail. This should be the case for the Kinships dataset used in [1,6,10] for instance ([1] showed that 3-way interactions are crucial there). Our submission shows that for modeling generic large-scale KB, one should rather model properly the most frequent connectivity patterns, which can be done with 2-way interactions only.


** Constraining the L2 norm of the embeddings of entities to be 1**
R4/R5: "How important is it to constrain the L2 norm of the embeddings of entities to be 1?" "Do any of the baseline models use a similar constraint?"

This constraint is important for our model, as it is for embedding methods learning a similarity function (SE, SME, LFM). This prevents the training to artificially satisfy the ranking criterion by blowing up entity embeddings norms. RESCAL (based on reconstruction) does not have this constraint.


** Other points **
- R4: "The paper really should report the hyperparameter values that produced the best performing models along with the corresponding training times and the parameter counts. It is also unclear whether L1 or L2 distances were used in the experiments."

On WN: k=20, learning rate=0.01, similarity=L1 distance, margin=2.
On FB15k: k=50, learning rate=0.01, similarity=L1 distance, margin=1.
On FB1M: k=50, learning rate=0.01, similarity=L2 distance, margin=1.

Training times were not provided because experiments have been conducted on different computers, making time comparison quite meaningless.


- R5: "Why do the numbers for SME not match those from [2] on WordNet? Was a different setup used?"

The authors of [2] published an erratum on their website (https://www.hds.utc.fr/~bordesan/dokuwiki/doku.php?id=en:aaai11_erratum) , because they reported some bugs in their data sets (overlap between train and test sets). We used the fixed WN data version they propose and our numbers correspond to those of the erratum.


- R5: "Why was the Neural Tensor Model not included in the experimental evaluation?"

The Neural Tensor Model has been published recently (ICLR'13 in May), shortly before the NIPS deadline. Besides, the code is not yet available, to the best of our knowledge.


- R6:"They also do not say how the latent dimensionality constant was chosen (268)."

We agree that the corresponding sentence is unclear. The dimension was chosen among {20, 50} on a validation set.

- R7:”The evaluation is ok, some more comparisons to recent work on tensor factorization would have made the paper better.”

We compare with six recent methods that have shown to reach state-of-the-art performances on various benchmarks in the last 2 years. This provides a wide coverage of what has been proposed recently. For us, the only missing method could be the Neural Tensor Model, for which we could not run the experiments (see reasons above).